# Tubulointerstitial Nephritis and Uveitis Syndrome (TINU): A Case Series in a Tertiary Care Uveitis Setting

**DOI:** 10.3390/jcm11174995

**Published:** 2022-08-25

**Authors:** Maria Pia Paroli, Daniele Cappiello, Davide Staccini, Rosalba Caccavale, Marino Paroli

**Affiliations:** 1Eye Clinic, Department of Sense Organs, Sapienza University of Rome, 00185 Rome, Italy; 2Division of Clinical Immunology, Department of Clinical, Anesthesiologic and Cardiovascular Sciences, Sapienza University of Rome, 00185 Rome, Italy

**Keywords:** tubulointerstitial nephritis, uveitis, TINU

## Abstract

Background: Tubulointerstitial nephritis and uveitis syndrome (TINU) is a rare disorder typically characterized by sudden-onset non-granulomatous anterior uveitis associated with tubulointerstitial nephritis (TIN). However, the prevalence and clinical features of TINU are still a matter of debate. To add information about TINU, we describe here the clinical features of a series of patients affected by TINU in a retrospective study. Methods: A total of 9358 clinical records of both adult and pediatric patients up to 21 years of age, referred to the Uveitis Center of the Sapienza University of Rome, were examined. The medical records covered a period from 1990 to 2020. Various demographic and clinical features were analyzed in patients who met the criteria for TINU. Results: Twenty-one patients with TINU were identified. TINU was classified as definite, possible, or probable by the currently recognized international criteria. The median age at diagnosis was 14 years (interquartile range 12–35). Females were predominant (15/21, 71.4%). In most cases (14/21, 66.6%), patients developed ocular disease concurrently with renal disease. The most frequent type of ocular involvement was bilateral anterior uveitis (9/21, 42.8%). In two cases, patients presented with bilateral intermediate uveitis; in three cases, they presented with bilateral or unilateral alternating posterior uveitis; and in four cases, they presented with bilateral panuveitis. In one case, the uveitis was anterior in the right eye (OD) and posterior in the left eye (OS), and two cases presented with bilateral asynchronous or unilateral alternating anterior uveitis. All patients received treatment with systemic corticosteroids and topical ocular therapy. At the end of the follow-up, a significant improvement in ocular signs and symptoms with a return to normal visual acuity was generally observed. In all patients, acute kidney injury (AKI) reverted completely and none progressed to chronic kidney disease (CKD). Conclusions: Patients with TINU may often present with atypical uveitis. We suggest that patients with sudden-onset uveitis, even if not bilateral anterior, should be referred to a nephologist for an assessment of the possible presence of renal disease.

## 1. Introduction

Tubulointerstitial nephritis and uveitis syndrome (TINU) is a rare disorder reported for the first time by Robert Dobrin et al. in 1975 [1]. A study by Mandeville et al. in 2001 first described the main features of the syndrome [2]. These consist of bilateral anterior uveitis with an onset less than 2 months before or 12 months after the onset of interstitial nephritis, in the absence of any other causes otherwise explaining eye or kidney pathology. The TINU diagnosis rate has been reported to reach 2% in specialized uveitis centers. In pediatric patients with characteristic sudden-onset bilateral uveitis, the diagnosis of TINU is made in about one-third of patients [2,3,4,5].

In recent years, TINU cases with ocular manifestations at presentation that are different from the original description have been often reported [3,6,7,8,9]. This suggests that early classification might underestimate the real prevalence of TINU, and that the diagnosis is likely to be more complex than previously believed. According to a recent systematic review, about 600 cases of TINU have been described in the scientific literature to date [10].

In the present paper, we report the demographic and clinical characteristics of a series of patients with TINU referred to our tertiary care uveitis center from 1990 to 2020 to provide further information on the clinical and epidemiological features of this syndrome.

## 2. Materials and Methods

Patient data were extracted from medical records in a tertiary care uveitis center at the Sapienza University of Rome. The study was approved by the local ethical committee and carried out in accordance with the Declaration of Helsinki principles for medical research. All patients included in this study gave written consent.

A total of 9358 medical records of patients with uveitis covering the period from 1990 to 2020 were reviewed. Of these, 734 were considered pediatric patients. The latter were defined as such on the basis of an age at the time of presentation of less than 20. Patients presenting with a suspected diagnosis of sarcoidosis, Sjogren syndrome, systemic lupus erythematosus, Behçet disease, tuberculosis, or syphilis were excluded from the study. Data on gender and age at the time of the diagnosis of the TINU patients were then collected. The uveitis was classified as anterior, intermediate, posterior, or panuveitis depending on which part of the uvea was primarily affected. A flow diagram showing the screening procedure followed in our center to identify patients meeting the diagnostic criteria for TINU is shown in Figure 1.

Other associated ocular findings, including cystoid macular edema (CME), papilledema, peripheral vasculitis, peripapillary choroidal neovascularization (CNV), and multifocal choroiditis, were also recorded. Tubulointerstitial nephritis (TIN) was diagnosed in 6 cases by kidney biopsy and in the remaining cases by typical acute kidney injury (AKI) presentation, characterized by abnormal renal function, an abnormal urinalysis, and systemic illness lasting ≥2 weeks according to the recognized international clinical criteria. Data on kidney function were collected in collaboration with the renal center of our institution [2,11,12]. The eGFR was calculated according to the Chronic Kidney Disease Epidemiological Collaboration (CKD-EPI) formula [13] The presence of autoantibodies, including ANA and/or ANCA, was also investigated.

Patients were then classified as having definite, probable, or possible TINU according to the criteria described by Mandeville et al. [2]. In more detail, TINU was classified as definite in the presence of bilateral anterior (typical) uveitis and either a positive biopsy or complete TIN; as probable in the presence of non-typical uveitis and a positive kidney biopsy or of anterior uveitis and incomplete TIN criteria; and as possible in the presence of atypical uveitis and incomplete TIN criteria. The median follow-up period was 12 months, from a minimum of 1 month to a maximum of 300 months. All patients were treated with systemic corticosteroids in addition to local eye therapy. The best corrected visual acuity (BCVA) was tested by examination with a Snellen eye chart.

Statistical analysis was performed using GraphPad Prism, version 8.0.0 for Windows, GraphPad Software, San Diego, CA, USA.

## 3. Results

Twenty-one patients met the criteria for the diagnosis of TINU. Of these, fifteen were female and six were male with an F:M ratio of 2.5:1. The median age at diagnosis was 14 years (interquartile range: 12–35) with a minimum age of 6 years and a maximum age of 62. The median age at diagnosis was higher in females (17.50, interquartile range of 13–44) than in males (12.50, interquartile range 10–16.25) with an F:M ratio of 1.4:1.

Six patients were adult and fifteen were pediatric. Ten patients were diagnosed with definite, ten with probable, and one with possible TINU. The concomitant presentation of renal symptoms and ocular symptoms was observed in the majority of cases (14/21, 66.7%). In four patients (19%), the first manifestation of the disease was TIN, while in three (14.3%), uveitis preceded the renal disease. The most common type of uveitis found was bilateral anterior uveitis (9/21). The most frequent type of uveitis was bilateral anterior (9/21, 42.8%), whereas two cases presented with bilateral intermediate and three cases presented with posterior—either bilateral or unilateral alternating. In four cases, the patients presented with bilateral panuveitis. In one case, uveitis was anterior in the right eye (OD) and posterior in the left eye (OS), whereas in two cases, the patients presented with bilateral asynchronous or unilateral alternating anterior uveitis. Uveitis bilaterality was significantly more frequent in adult patients. We also analyzed the ocular pathologies associated with uveitis. CME was detected in one case (Subject 1), peripheral vasculitis in three cases (Subjects 1, 6, and 13), multifocal choroiditis in one case (Subject 2), papilledema in four cases (Subjects 6, 9, 12, and 13), and CNV in one case (Subject 6).

Regarding renal disease, TIN was idiopathic in thirteen patients (61.9% of cases), was related to the use of antibiotics in four patients (19%), was followed by upper airways infection in three patients (14.3%), and in one case the TIN was of possible autoimmune etiology due to the presence of ANA in the serum (subject 1).

The treatment of all patients consisted of systemic corticosteroids in combination with mydriatics/cycloplegics. The remission of AKI without progression to chronic kidney disease (CKD) was obtained in all cases. In 13 patients (61.9%), systemic corticosteroids were administered in addition to local therapy, even after the renal symptoms disappeared, to achieve the complete remission of ocular inflammation as well. The treatment resulted in restored visual acuity in all patients (BCVA of 20/20 in all the patients at follow up). During the follow-up period, 12 patients (57.1%) had at least one episode of uveitis flare-up, which was successfully treated with standard therapy.

The demographic and TINU characteristics, the most relevant features of the kidney function at the time of the renal disease diagnosis, and the TINU-related clues in the medical history of individual patients are summarized in Table 1. The patients were numbered progressively according to the period in which they were referred to our center. A comparison of the results obtained in the adult and pediatric populations are shown in Table 2. A *p*-value < 0.05 was considered significant. Fisher’s exact test was used to compare the categorical variables.

## 4. Discussion

In the present paper, we retrospectively analyzed a series of patients affected by TINU that were referred to a tertiary care uveitis center.

We found a total of 21 cases of TINU after reviewing 9358 medical records of patients with suspected or established uveitis. Patients with TINU accounted for 0.2% of all the patients admitted from 1990 to 2020. When we considered only patients with an age up to 21 years from a subgroup of 734 pediatric records, we found that patients with TINU were 15/21, thus accounting for 2% of the patients referred for pediatric uveitis. It has been reported that the syndrome comprises approximately 1–2% of uveitis practice, but it may account for nearly 32% of pediatric patients with characteristic sudden onset bilateral anterior uveitis [2,3,4]. A recent systematic review identified a total of 592 cases as reported in the scientific literature to date [10]. The prevalence of patients with TINU described in this study is lower than that reported in previous studies. However, we should consider that the calculation was made on medical records that included some patients referred for suspected uveitis, but who were eventually diagnosed as having other ocular diseases.

In our case series, the F:M ratio of patients with TINU was 2.5:1. In pediatric patients, this ratio was 2:1, confirming the predominance of females in this syndrome. A female predominance of about 3:1 has been previously reported [2,3]. However, this gender effect appeared weaker in other studies [14,15,16]. As TINU is a multisystemic condition, autoimmune mechanisms might be relevant to the onset of the syndrome in predisposed individuals, thus explaining the prevalence in females [17,18,19]. Our data showed that the renal and ocular damage appeared simultaneously in most cases. This finding is in agreement with previous studies [2,3,20]. In the pediatric subgroup, this finding was also confirmed.

Regarding the type of uveitis, bilateral anterior uveitis was the most frequently observed. However, a significant number of patients presented with intermediate, posterior, or even panuveitis. This finding is in agreement with recent reports that have challenged early classification [6,7,9]. We also observed that additional ocular pathologies may accompany uveitis in several cases. These included CME, papilledema, peripheral vasculitis, multifocal choroiditis, and peripapillary CNV. In this regard, posterior atypical findings are being described in an increasing number of TINU cases [21,22,23,24,25].

In some cases, TINU was precipitated by drugs (antibiotics) or infections of the upper airways, but in most cases no trigger was identified. It has been reported that drugs and infections by several pathogens may induce TINU [26]. However, most studies have been retrospective, and it has been noticed that a substantial proportion of TINU risk factors are common in the general population and may even coexist [16].

All patients were treated with systemic steroids and topical eye therapy. The progression from AKI to CKD was not observed in any patient with TINU. It has been observed that the progression to CKD mostly occurs in adults. However, it has been suggested that the evolution toward CKD might require partially premorbid conditions of the kidneys, possibly dependent on lifestyle and comorbidities [10]. Patients in our case series were likely not to have risk factors for the progression to CKD. Uveitis also had a good prognosis in our patients, with the recovery of pre-TINU visual acuity in almost all the patients as previously reported [3]. However, 12 patients had at least one episode of uveitis flare-up. The ocular disease has been reported to recur in up to 50% of patients after corticosteroid withdrawal [2]. Therefore, the prolonged follow-up of these patients is required.

Our study has some limitations. Firstly, it was a single-center study that analyzed a relatively small number of patients. However, it should be considered that TINU is a rare syndrome, and it is not easy to collect many cases. Likely, several differences were not significant because of the small sample size. Multicenter studies will therefore be necessary to obtain more reliable results. A multi-institutional study in pediatric patients is also needed. Second, patients with simultaneous or sequential symptomatic bilateral anterior uveitis that were referred to our center were not routinely tested for urinary beta-2-microglobulin. The elevation of this marker has emerged as a sensitive noninvasive test that is useful in the diagnosis of TINU [5,9]. Therefore, it cannot be ruled out that a significant number of patients remained undiagnosed. Third, the retrospective design of this study raises the possibility of a recording bias as well as the likelihood of missing data.

In conclusion, the data obtained from our study underline the complexity of the epidemiology, clinical presentation, and diagnosis of TINU. We suggest that any patient presenting not only with acute bilateral uveitis, but also with atypical uveitis should undergo a thorough nephrological evaluation. On the other hand, patients with TIN should be screened for the presence of uveitis. The strict collaboration among ophthalmologists, pediatricians, and nephrologists is the key factor for the optimal management of patients with this rare syndrome.

## Figures and Tables

**Figure 1 jcm-11-04995-f001:**
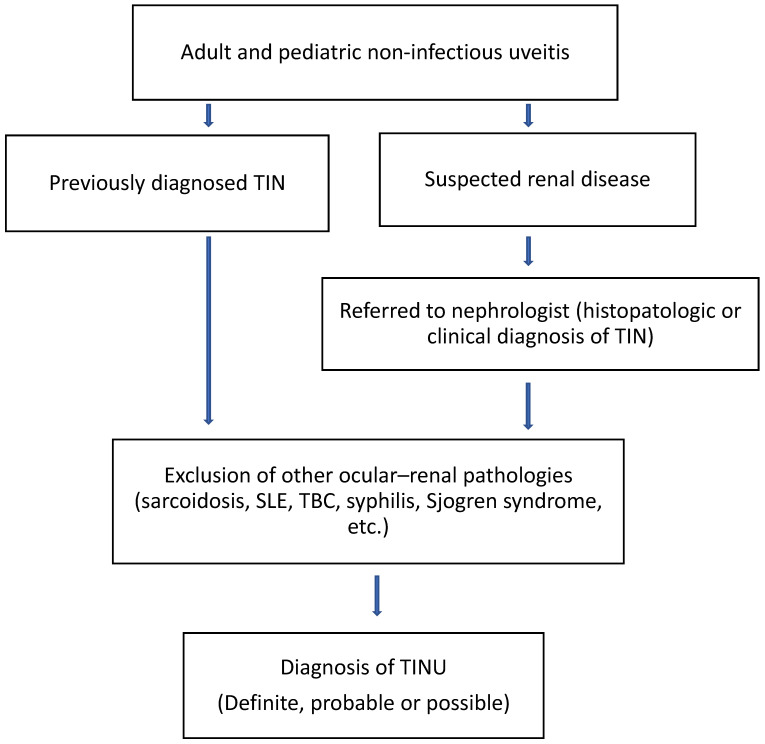
Flow chart showing the screening procedure followed to identify patients meeting the diagnostic criteria of TINU. TIN, tubulointerstitial nephritis, SLE, systemic lupus erytematosus, TBC, tuberculosis, TINU, tubulointerstitial nephritis and uveitis syndrome.

**Table 1 jcm-11-04995-t001:** Demographic and clinical characteristics of patients. TINU, tubulointerstitial nephritis and uveitis syndrome, eGFR, estimated glomerular filtration rate.

Subject	Age at Diagnosis	Gender	TINU-Relevant Medical History	Serum Creatinine Level (mg/dL)	eGFR ^1^	Organ Involved First (Kidney/Eye)	TINU	Uveitis Type ^2^	Uveitis Laterality ^3^	Biopsy ^4^
1	42	Female	Immune-related *	1.3	25	Eye	Probable	Intermediate	OU	Positive
2	23	Male	Viral infection	1.2	38	Concurrent	Probable	Panuveitis	OU	ND
3	16	Female	None	1.2	45	Concurrent	Probable	OD anterior OS posterior	OU	ND
4	45	Female	Viral infection	1.8	32	Concurrent	Definite	Anterior	OU	ND
5	62	Female	Antibiotics	2.4	24	Concurrent	Definite	Anterior	OU	ND
6	6	Female	Antibiotics	1.8	26	Kidney	Probable	Panuveitis	OU	ND
7	10	Male	None	1.4	44	Concurrent	Definite	Anterior	OU	ND
8	19	Female	None	2.6	42	Concurrent	Definite	Anterior	OU	Positive
9	19	Female	None	1.6	32	Kidney	Probable	Posterior	OD-->OS	Positive
10	14	Female	None	1.6	26	Concurrent	Definite	Anterior	OU	Positive
11	13	Male	Antibiotics	5.7	20	Concurrent	Definite	Anterior	OU	Positive
12	44	Female	None	9	50	Eye	Probable	Panuveitis	OU	Positive
13	12	Female	None	2.2	28	Concurrent	Probable	Posterior	OU	ND
14	14	Female	Viral infection	1.4	44	Eye	Possible	Intermediate	OU	ND
15	14	Male	None	1.3	52	Concurrent	Probable	Anterior	OS-->OU	ND
16	10	Male	None	1.4	44	Concurrent	Definite	Anterior	OU	ND
17	13	Female	None	1.3	38	Concurrent	Definite	Anterior	OU	ND
18	12	Male	None	2.4	21	Kidney	Possible	Panuveitis	OU	ND
19	48	Female	None	1.8	32	Concurrent	Definite	Anterior	OU	ND
20	17	Female	None	1.6	45	Concurrent	Probable	Anterior	OD-->OD	ND
21	13	Female	Antibiotics	2.1	34	Kidney	Probable	Posterior	OS-->OD	ND

^1^ = mg/mL/1.73 m^2^; ^2^ OD = right eye; OS = left eye; ^3^ OU = bilateral simultaneous; OS-->OU = bilateral asynchronous; OD-->OS and OS-->OD = unilateral alternating; ^4^ ND = not done; * = ANA+.

**Table 2 jcm-11-04995-t002:** Differences in uveitis characteristics of adult and pediatric patients.

Characteristics	≥21 Years *n* = 6	≤21 Years *n* = 15	*p*-Value
Gender (female), *n* (%)	5 (83.3%)	10 (66.7%)	NS
Uveitis, *n* (%)			
1. Anterior	3 (50%)	9 (60%)	NS
2. Posterior	0 (0%)	3 (20%)	NS
3. Intermediate	1 (16.7%)	1 (6.7%)	NS
4. Panuveitis	2 (33.3%)	2 (13.3%)	NS
Uveitis (unilateral:bilateral), *n* (%)	6 (100%)	4 (26.7%)	*p* < 0.001
Uveitis onset, *n* (%)			
1. Before TIN	2 (33.3%)	1 (6.7%)	NS
2. Concurrent with TIN	4(66.7%)	10 (66.7%)	NS
3. After TIN	0 (0%)	4 (26.6%)	NS

NS = not significant.

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
