# Peer review of "Tubulointerstitial Nephritis and Uveitis Syndrome (TINU): A Case Series in a Tertiary Care Uveitis Setting"

_jcm, 2022, doi:10.3390/jcm11174995_

Round 1
Reviewer 1 Report
This study aimed to address the demographic and clinical characteristics of patients presenting with a rare disorder. The study is interesting, but as a nephrologist, I have some additional comments:
1. In the absence of a kidney biopsy, the diagnosis of tubulo-interstitial nephritis is uncertain. Elevations in serum creatinine or abnormal urine analysis may indicate an acute kidney injury of multi-factorial etiology.
2. In Table 1, the authors can provide some additional data for their patients (more detailed medical history, serum creatinine levels and eGFR), if these data are available.
3. A flow diagram to describe how did you screen the patients to identify and include those meeting the diagnostic criteria of TINU syndrome could be helpful.
4. The retrospective design of this study raises the possibility of recording bias as well as the likelihood of missing data. This limitation needs to be acknowledged.
Author Response
Reviewer#1.
- Query: In the absence of a kidney biopsy, the diagnosis of tubulo-interstitial nephritis is uncertain, and elevations in serum creatinine or abnormal urine analysis may indicate an acute kidney injury of multi-factorial etiology. Answer: We agree on this point. Biopsy was performed in 6 out of 21 of our patients. In the remaining patients, diagnosis was made on the basis of recognized international clinical criteria. We have underlined the issue in the text and added the relative reference
2) Query: In Table 1, the authors can provide some additional data for their patients (more detailed medical history, serum creatinine levels and eGFR), if these data are available. Answer: Three columns were added to table 1 concerning TINU-related medical history, serum creatinine level and eGFR values at the time of TIN diagnosis. We also added a commentary within the text and a bibliography reference regarding the method we used to calculate eGFR
3) Query: A flow diagram to describe how did you screen the patients to identify and include those meeting the diagnostic criteria of TINU syndrome could be helpful. Answer: A flow diagram showing the screening procedure we followed to identify patients meeting the diagnostic criteria of TINU syndrome has been added. This flow diagram is now figure 1.
4) Query: The retrospective design of this study raises the possibility of recording bias as well as the likelihood of missing data. This limitation needs to be acknowledged. Answer: This study limitation has been acknowledged within the text in the Discussion section.
Reviewer 2 Report
This is a manuscript that analyzes the clinical characteristics of TINU patients by picking up patients with TIN from uveitis patients in single institution for past about 30 years. I think your data is reliable because they are targeting a large number of uveitis patients. However, as you also state, the number of TINU patients may be higher because TIN may not have been properly diagnosed in the past. If you search in cooperation with a facility with a renal center, it seems that the data became a little more interesting. What interested me about your manuscript was the data that in children, uveitis is often recognized after the onset of TIN. Unfortunately, in adults, the number of cases is too small to be definitive. As a pediatric nephrologist, I found the data interesting. I would like to proceed with a multi-institutional study.
Author Response
Reviewer#2.
- Query: If you search in cooperation with a facility with a renal center, it seems that the data became a little more interesting. Answer: In cooperation with the renal center of our institution, further data on kidney function for diagnosis of tubulointerstitial nephritis have now been added to table 1.
- Query: As a pediatric nephrologist, I found the data interesting. I would like to proceed with a multi-institutional study. Answer: The need for a multi-institutional study in pediatric patients has been underlined in the Discussion section.